# Anti-TNF (adalimumab) injection for the treatment of pain-predominant early-stage frozen shoulder: the Anti-Freaze-Feasibility randomised controlled trial

Sally Hopewell [1] Cynthia Srikesavan,[2,3] Alison Evans,[4] Fema Er,[1] Amar Rangan [5,6] Jane Preece,[7] Anne Francis,[1] M Sofia Massa [1] Marc Feldmann,[4] Sarah Lamb,[3] Jagdeep Nanchahal[4]

For numbered affiliations see end of article.

**Correspondence to**
Dr Sally Hopewell;
sally.hopewell@csm.ox.ac.uk

## ABSTRACT

**Objective** The Anti-Freaze-F (AFF) trial assessed the feasibility of conducting a definitive trial to determine whether intra-articular injection of adalimumab can reduce pain and improve function in people with pain-predominant early-stage frozen shoulder.

**Design** Multicentre, randomised feasibility trial, with embedded qualitative study.

**Setting** Four UK National Health Service (NHS) musculoskeletal and related physiotherapy services.

**Participants** Adults ≥18 years with new episode of shoulder pain attributable to early-stage frozen shoulder.

**Interventions** Participants were randomised (centralised computer generated 1:1 allocation) to either ultrasound-guided intra-articular injection of: (1) adalimumab (160 mg) or (2) placebo (saline (0.9% sodium chloride)). Participants and outcome assessors were blinded to treatment allocation. Second injection of allocated treatment (adalimumab 80 mg) or equivalent placebo was administered 2–3 weeks later.

**Primary feasibility objectives** (1) Ability to screen and identify participants; (2) willingness of eligible participants to consent and be randomised; (3) practicalities of delivering the intervention; (4) SD of the Shoulder Pain and Disability Index (SPADI) score and attrition rate at 3 months.

**Results** Between 31 May 2022 and 7 February 2023, 156 patients were screened of whom 39 (25%) were eligible. The main reasons for ineligibility were other shoulder disorder (38.5%; n=45/117) or no longer in pain-predominant frozen shoulder (33.3%; n=39/117). Of the 39 eligible patients, nine (23.1%) consented to be randomised (adalimumab n=4; placebo n=5). The main reason patients declined was because they preferred receiving steroid injection (n=13). All participants received treatment as allocated. The mean time from randomisation to first injection was 12.3 (adalimumab) and 7.2 days (placebo). Completion rates for patient-reported and clinician-assessed outcomes were 100%.

**Conclusion** This study demonstrated that current NHS musculoskeletal physiotherapy settings yielded only small numbers of participants, too few to make a trial viable.

## STRENGTHS AND LIMITATIONS OF THIS STUDY

⇒ The Anti-Freaze-F study was a pilot randomised controlled trial with an embedded qualitative study to test the effectiveness of a new treatment for people with early stage, pain-predominant frozen shoulder.

⇒ Participants were randomised (centralised computer generated 1:1 allocation) to either ultrasound-guided intra-articular injection of adalimumab or placebo injection. Participants and outcome assessors were blinded to treatment allocation.

⇒ We assessed the feasibility of conducting a large multicentre randomised trial in terms of the ability to screen and identify participants, their willingness to consent and be randomised, the practicalities of delivering the intervention and attrition rates at 3 months.

⇒ While all nine participants recruited into the trial received both their first and second injections within the specific timeframe, it is unclear whether this would be same for larger definitive trial.

⇒ Similarly, while all participants completed both patient-reported and clinician-assessed outcomes at 3 months, it is unlikely this would be same for larger trial with longer follow-up.

This was because many patients had passed the early stage of frozen shoulder or had already formulated a preference for treatment.

**Trial registration number** ISRCTN 27075727, EudraCT 2021-03509-23, ClinicalTrials.gov NCT05299242 (REC 21/NE/0214).

## INTRODUCTION

Frozen shoulder (adhesive capsulitis) is a common and extremely painful and debilitating condition. Affected individuals struggle with activities of daily living and significant sleep disturbances as a result of severe pain.[1]

It affects about 9% of people in the United Kingdom (UK) aged 25–64 years,[2] and 20% develop the same problem in the other shoulder.[3] Frozen shoulder may develop as a primary condition or secondarily following surgery or trauma.

The classic description of the development of frozen shoulder is of three overlapping phases.[4 5] The initial pain-predominant inflammatory phase is characterised by constant pain and difficulty sleeping and lasts between 3 and 9 months. This progresses to a stiffness-predominant fibrotic phase, with progressive restriction of motion, particularly external rotation and elevation of the shoulder, and impairment of function and lasts between 4 and 12 months. The pain changes from being constant to being manifest at the end of range of motion and of reduced intensity. Over the final phase, there is a gradual improvement in range of motion and stiffness over a 12 to 48-month period, although pain may persist with end-range movements. The average duration of the condition is 30 months (range 1–3.5 years) although full resolution of symptoms does not always occur.[6]

The aetiology of frozen shoulder is poorly understood and consequently there is no consensus on the optimal treatment. The majority of patients with early-stage pain-predominant frozen shoulder are managed in primary care or at primary care interface musculoskeletal services by physiotherapists and general practitioners. During this stage, standard treatment consists of rest, advice, analgesics, physiotherapy and corticosteroid injections to address the symptoms. Current treatments available to patients with frozen shoulder are of limited efficacy. Two Cochrane reviews concluded that while oral steroid or local steroid injections lead to short-term benefit in pain and range of motion, the effects are not maintained beyond 6 weeks.[7 8] A more recent systematic review (based on data from five randomised trials) also found some benefit of corticosteroid injection compared with placebo, but this pain relief was not sustained in the long term.[9] Other Cochrane reviews concluded that there is no evidence that physiotherapy or ultrasound therapy are beneficial,[10] and that manual therapy with exercise is less effective than corticosteroid injection in the short term.[11] The UK FROST compared the effects of physiotherapy plus corticosteroid injection, manipulation under anaesthesia with a steroid injection, and arthroscopic capsular release supplemented with manipulation.[12] None of the treatments was found to be clinically superior.

Our study (Anti-Freaze-F) was designed to specifically target people with early-stage pain-predominant frozen shoulder. We assessed the feasibility of conducting a large multicentre randomised trial to test whether giving an intra-articular injection of adalimumab (a drug targeting the inflammatory mediator tumour necrosis factor (TNF)) can reduce pain and prevent the condition from getting worse, if given during the early pain-predominant stage.

Approximately 50% of people with frozen shoulder also develop Dupuytren's disease.[13] Dupuytren's disease has a prominent genetic component, and in a study of twins, the heritability was estimated at 80%.[14] We have recently confirmed that there is a significant genetic correlation between Dupuytren's disease and frozen shoulder.[15] The pathology of the two conditions has several similarities.[16 17] A systematic review of the pathophysiology of frozen shoulder identified the presence of fibrosis and the role of inflammation.[6] The affected tissues are infiltrated by immune cells and there are elevated levels of proinflammatory cytokines, including TNF,[18 19] with myofibroblasts driving the development of fibrosis.[16] In a dose-ranging proof of concept phase 2a followed by a 2b clinical trial (RIDD trial), we found that injection of anti-TNF (adalimumab) directly into Dupuytren's nodules led to significant downregulation of the myofibroblast phenotype and reduction in nodule hardness (primary endpoint) and nodule size on ultrasound scan (secondary endpoint).[20 21]

## Objectives

The aim of the Anti-Freaze-F (AFF) trial was to assess the feasibility of conducting a large randomised controlled trial to assess whether an intra-articular injection of adalimumab (anti-TNF) can reduce pain and improve function in people with early-stage frozen shoulder. The primary objectives were to assess the:

► Ability to screen and identify potential participants with pain-predominant early stage frozen shoulder.
► Willingness of eligible participants to consent and be randomised to intervention.
► Practicalities of delivering the intervention, including time to first injection and number of participants receiving second injection.
► SD of the Shoulder Pain and Disability Index (SPADI) score and attrition rate at 3 months (ie, 12 weeks) postrandomisation in order to estimate the sample size for a definitive trial.

Secondary objectives were to assess the follow-up rates and viability of patient-reported outcome measures and clinician-assessed shoulder range of motion at 3 months (ie, 12 weeks).

The objectives of the embedded qualitative study were to explore the participant experience of being recruited to the AFF trial, the treatment received and follow-up schedule, and to understand what helps participant recruitment to the trial intervention.

## METHODS
### Study design
A multi-centre, randomised, double-blind, parallel group, feasibility trial with an embedded qualitative study.

### Setting
Participants were recruited from four National Health Service (NHS) musculoskeletal services and their related physiotherapy services, with treatment delivered within

these services or local secondary care dependant on the local service provision.

## Study participants

Participants were eligible if they were aged 18 years and above with a new episode of shoulder pain attributable to pain-predominant stage of frozen shoulder (ie, within approximately 3 to 9 months of onset of symptoms), which was diagnosed clinically using the criteria set out in the British Elbow and Shoulder Society (BESS) guidelines.[22] In line with the trial protocol, imaging, including plain radiographs, may be used to help confirm a diagnosis of frozen shoulder by excluding other pathology such as glenohumeral arthritis as per standard NHS care. However, is not always necessary and hence routine radiology was not as part of the trial procedures.[23] Patients were excluded if they had frozen shoulder secondary to significant shoulder trauma; other shoulder disorders (eg, inflammatory arthritis, rotator cuff disorders, glenohumeral joint instability) or with red flags consistent with the criteria set out in the BESS guidelines[22]; bilateral early stage frozen shoulder; had received corticosteroid injection for shoulder pain in the last 12 weeks; were currently taking any anti-TNF drug, or being treated with coumarin anticoagulants, such as warfarin; had significant renal or hepatic impairment, or had contraindications to anti-TNF injection. Detailed inclusion and exclusion criteria are described in the trial protocol.[23] Patients who met the eligibility criteria and wanted to participate were approached for written informed consent by a research facilitator trained in Good Clinical Practice at each participating site. All participants underwent serological testing to check for latent tuberculosis (TB) and hepatitis B surface antigen. The blood tests were performed during the baseline assessment or at the time the participant attended for their first injection appointment (depending on the local site provision). The risks of reactivation following a single injection are low. If any participants had shown a positive serology test result, they would have been referred to their local infectious disease service and would not have received the second injection.

## Randomisation

Consented participants were randomly assigned to receive either: (1) intra-articular injection of adalimumab or (2) placebo injection (saline (0.9% sodium chloride), both under ultrasound guidance. Randomisation (1:1) was done by the research facilitator at each site using the centralised randomisation service provided by the Oxford Clinical Trials Research Unit once the patient was enrolled and baseline assessment and questionnaire were completed. Randomisation was computer generated and stratified by study site using a variable block size (variable block sizes of 2, 4 and 6 in a ratio of 1:2:1) to ensure the participants from each study site had equal chance of receiving either intervention.

## Blinding

Study participants and site staff, except pharmacy staff, were blinded to treatment allocation. The clinician delivering the treatment injection was not blinded but was not involved in further trial-specific assessment of the participant. The trial statistician and data entry personnel were not blinded to the treatment allocation. The remaining members of the trial management team, including the staff conducting the qualitative interviews, were blinded to treatment allocation until after data analysis was complete.

## Adalimumab/placebo injection

As per their treatment allocation, participants received either intra-articular injection of adalimumab (160 mg in 3.2 mL for the first injection, 80 mg in 1.6 mL for second injection) or placebo (normal saline (0.9% sodium chloride) 3.2 mL for the first injection and 1.6 mL for the second injection). Full details of the injection delivery have been reported previously[23] and are described here in brief.

Appointments were coordinated, so that participants received their first injection within approximately 2 weeks of randomisation, and second injection 2–3 weeks after the first (unless the participant declined, tested positive for TB or hepatitis B surface antigen or had a related grade 3 or above adverse event). The injection was given into the anterior shoulder joint space in the rotator cuff interval where there is maximal inflammation of the capsule and synovium,[24] under guided ultrasound by an appropriately qualified practitioner.

The adalimumab/placebo injection was dispensed by the local site pharmacy and sealed in identical sized and sealed opaque packaging. Preparation of the adalimumab/placebo injection took place in a clinic room/area separate from the participant to ensure the participant remained blinded to treatment allocation. Both adalimumab and placebo have a similar appearance; therefore, the two treatments are indistinguishable. The skin at the site of injection of adalimumab/placebo was infiltrated with local anaesthetic to reduce the pain of the injection as per local practice. Injection details, including success of the intra-articular injection (ie, fully or partial), were recorded on a trial-specific injection treatment log.

All participants received a written physiotherapy advice leaflet providing education and advice about frozen shoulder and pain management.[4] The advice leaflet also included simple self-guided exercises, which participants could use to increase shoulder joint movements, once the early pain-predominant stage reduced. Participants were advised they may seek other forms of treatment during the follow-up period of the trial but were informed that they should use usual routes (eg, through GP referral) to do so. Participants were also advised that they could seek corticosteroid injection but were asked to wait until after the 3-month follow-up period.

## Outcome measures

### Feasibility success criteria

To determine the feasibility of a definitive randomised controlled trial, the prespecified success criteria were:

► Feasibility to recruit: ≥33% of potentially eligible patients with frozen shoulder screened eligible for recruitment.

► Success of consent process: ≥33% of eligible participants consented.

► Intervention delivery: ≥75% of participants receive first injection as randomised within specified timeframe.

In addition, we collected outcomes at baseline and at 3 months to assess the feasibility of collecting these in a future definitive trial and to obtain the variability estimates required for estimation of the sample size of a definitive trial. Patient-reported outcomes, collected via online/paper questionnaire, included shoulder pain and function measured using the SPADI scale (primary outcome for definitive trial);[25][26] subdomains of pain (SPADI 5-item pain subscale), function (SPADI 8-item disability subscale)[25][26]; shoulder range of motion (Participant Shoulder Movement Questionnaire); psychological factors (Fear-Avoidance Belief Questionnaire)[27]; pain self-efficacy questionnaire[28]; sleep disturbance (Insomnia Severity Index)[29]; patient global impression of change[30]; return to desired activities; and additional health resource use. At baseline and the 3-month face-to-face follow-up appointment, a blinded assessor measured the shoulder movements including active flexion, extension, abduction internal and external rotations, using a manual goniometer.

## Adverse events

The safety profile of adalimumab is well known, with the most common adverse reactions being mild injection site reactions. The Common Terminology Criteria for Adverse Events (CTCAE) V.5.0 was used to guide recording of any adverse events including grading of the event. Only clinician-assessed adverse events, graded 3 and above, occurring during the trial for each participant, from their consent until the 3-month follow-up, related to the trial medication (adalimumab/placebo) were recorded.

## Sample size

The main feasibility objective and, therefore, the basis of the sample size estimate were participant recruitment per centre. The target sample size was 84 participants, equivalent of 1 to 2 participants per month per site over 12 months. Seventy is the recommended minimum target sample size when including an estimate of the SD in an external pilot trial.[31] The sample size was increased to 84 to increase precision of the estimate of the SD of SPADI at 3 months, the proposed primary outcome for the definitive trial and to consider possible attrition based on an attrition rate of 15%.

## Statistical analysis

Feasibility outcomes were reported as numbers and percentages. Baseline characteristics and outcome measures were reported using descriptive statistics, separately per group and overall, using either the mean and SD for continuous variables and number and percentage of participants in each group for categorical variables.

## Embedded qualitative study

We planned to interview a purposive sample of up to 15 participants (or until we reach data saturation) to provide variability for age, gender, ethnicity and geographical representation. A qualitative researcher (CS), blinded to treatment allocation, conducted the telephone interviews using a semistructured interview guide with open-ended questions. Interviews were audio-taped and transcribed verbatim. Data were analysed following Braun and Clarke's thematic analysis method.[32] Interviews were coded using NVivo qualitative software V.12 and categorised into themes. Participant narrative quotes were used to illustrate the themes.

## Patient and public involvement

Patient representatives were involved in the design, conduct and reporting of study, including reviewing of patient-facing study materials (eg, patient information sheet, physiotherapy advice booklet and patient follow-up questionnaires) and through involvement of the trial management and trial oversight committees.

## RESULTS

Regulatory approvals were obtained on 22 December 2021. Patient screening and enrolment began 31 May 2022 and ended 7 February 2023; completion of the 3-month outcome assessment and follow-up ended 9 May 2023. Delays in obtaining site approvals due to the continuing impact of COVID-19 on staffing levels meant that site opening and thus patient screening was delayed, resulting in the recruitment period being just over 8 months as opposed to the planned 12 months. Delays in local approvals at sites also meant we were only able to open four of the planned five sites as stipulated in the trial protocol.

## Feasibility objectives

### Feasibility to recruit

Of the 156 patients screened, 39 (25%) were eligible for the trial. The main reasons screened patients were not eligible were because they had other shoulder disorders (eg, inflammatory arthritis, rotator cuff disorders, glenohumeral joint instability) (n=45/117; 38.5%); were no longer in the pain-predominant stage of frozen shoulder (n=39/117; 33.3%); had frozen shoulder due to significant shoulder trauma or other causes (eg, recent breast cancer surgery) (n=11/117; 9.4%); had bilateral early stage frozen shoulder (n=7/117; 6.0%) or had received

corticosteroid injection for shoulder pain within the last 12 weeks to either shoulder (n=5/117; 4.3%) (figure 1).

## Success of consent process

Of the 39 patients eligible for the trial, 29 (74.4%) declined to participate, the main reasons being they already had a treatment preference for receiving steroid injection (n=13/29; 44.8%), did not wish to be randomised (n=7/29; 24.1%) or already had a treatment preference for not receiving an injection (n=4/29; 13.8%). In addition, one patient who was eligible was not consented due to close of trial recruitment. This resulted in nine (n=9/39; 23.1%) eligible patients being consented into the trial (figure 1); four were randomised to receive adalimumab injection and five to placebo injection. Baseline characteristics are summarised in table 1. Participants had a mean age of 54.9 (SD 5.6) years and three (33.3%) of the nine participants were women; all were white British. The average (mean) duration of symptoms was 15.4 weeks (SD 9.0). The mean SPADI score at baseline (primary outcome for definitive trial) was 70.8 (SD 16.6) for the adalimumab group and 55.4 (SD 21.5) for the placebo group.

## Intervention delivery

All participants received their treatment as allocated and in line with the trial protocol. The mean time from randomisation to first injection was 12.3 days for the adalimumab group and 7.2 days for the placebo group. One participant in the adalimumab group received their injection 18 days post-randomisation as they were not available until then. The mean time from receiving first to second injection was similar for adalimumab (18.5 days) and placebo (20.8 days) (table 2).

## Patient retention and outcome measure data collection

Data for patient-reported outcomes and clinician-assessed range of shoulder motions are shown in online supplemental tables S1–S3. The completion rates at baseline and 3 months were 100% (n=9/9) for both patient-reported outcomes, completed by questionnaire, and clinician-assessed shoulder motion. Between-group difference were not calculated as only nine participants were recruited into the trial. Online supplemental figure S1 shows the change from baseline and at 3 months for shoulder pain and function (measured using the SPADI scale) per participant for adalimumab and placebo injection.

## Adverse events

No graded 3 or above, adverse events occurred during the trial related to the trial medication (adalimumab/placebo).

## Embedded qualitative study

Eight participants took part in the telephone interviews (four from each injection group), which included six men and two women with an average age of 55.5 years (range 46 to 62 years). The average duration of interviews

was 24 min (range 14 min–31 min). Five participants had received both injections at the time of interviews. Four key themes emerged: (1) experiences with pain-predominant stage of frozen shoulder; (2) perceptions about participating in the AFF trial; (3) perceptions about the trial processes and (4) perceptions about the effects of injections. Themes and illustrative quotes are presented in table 3.

## Experiences with pain-predominant phase of frozen shoulder

This theme describes the impact of the pain-predominant phase of the frozen shoulder. Most participants reported severe levels of pain, particularly with certain movements or during sleep. The shoulder pain was described as 'very acute', 'annoying', 'disruptive', 'excruciating' or 'completely unbearable'. Participants also discussed how their everyday function was limited (eg, difficulties with getting dressed or driving). Sleep was substantially affected by pain, with participants reporting shorter sleep times and poorer sleep quality. They frequently woke up in the night due to severe pain while changing position.

## Perceptions about participating in the trial

This theme describes the participants' views on participating in the AFF trial. Overall, participants were keen to find a solution to their shoulder problem. Severe pain, reduced function and sleep and previous treatments such as steroid injections or physiotherapy that were not helpful were other reasons that influenced their decision to take part. Participants also reported altruistic reasons to benefit others with similar problems. Their expectations on taking part were to get adalimumab injection, pain relief and improved function. Participants also expressed a clear understanding and acceptance of the randomisation process, mainly attributing to the trial-related information provided to them. Having a 50% chance of getting the placebo injection was a concern. However, participants discussed various reasons, including long waiting time for steroid injections, positive expectations about adalimumab and flexibility to access standard care (if AFF trial injections did not work) that helped them take a 'balanced view' in participating.

## Perceptions about trial processes

This theme describes participant views about trial processes, including provision of information, delivery of injections, outcome assessments and their overall experience. Participants described that all trial-related information such as randomisation and delivery of injections was communicated effectively, with opportunities to clarify their queries and make a decision. Injection appointments were scheduled quickly, and participants were well-informed and supported during the sessions. Many reported discomfort at the time of injections. Participants perceived the outcome measurements (clinician-assessed measurements and self-reports) as relevant to their condition and had no issues completing them. They also highlighted their positive experiences with the trial personnel

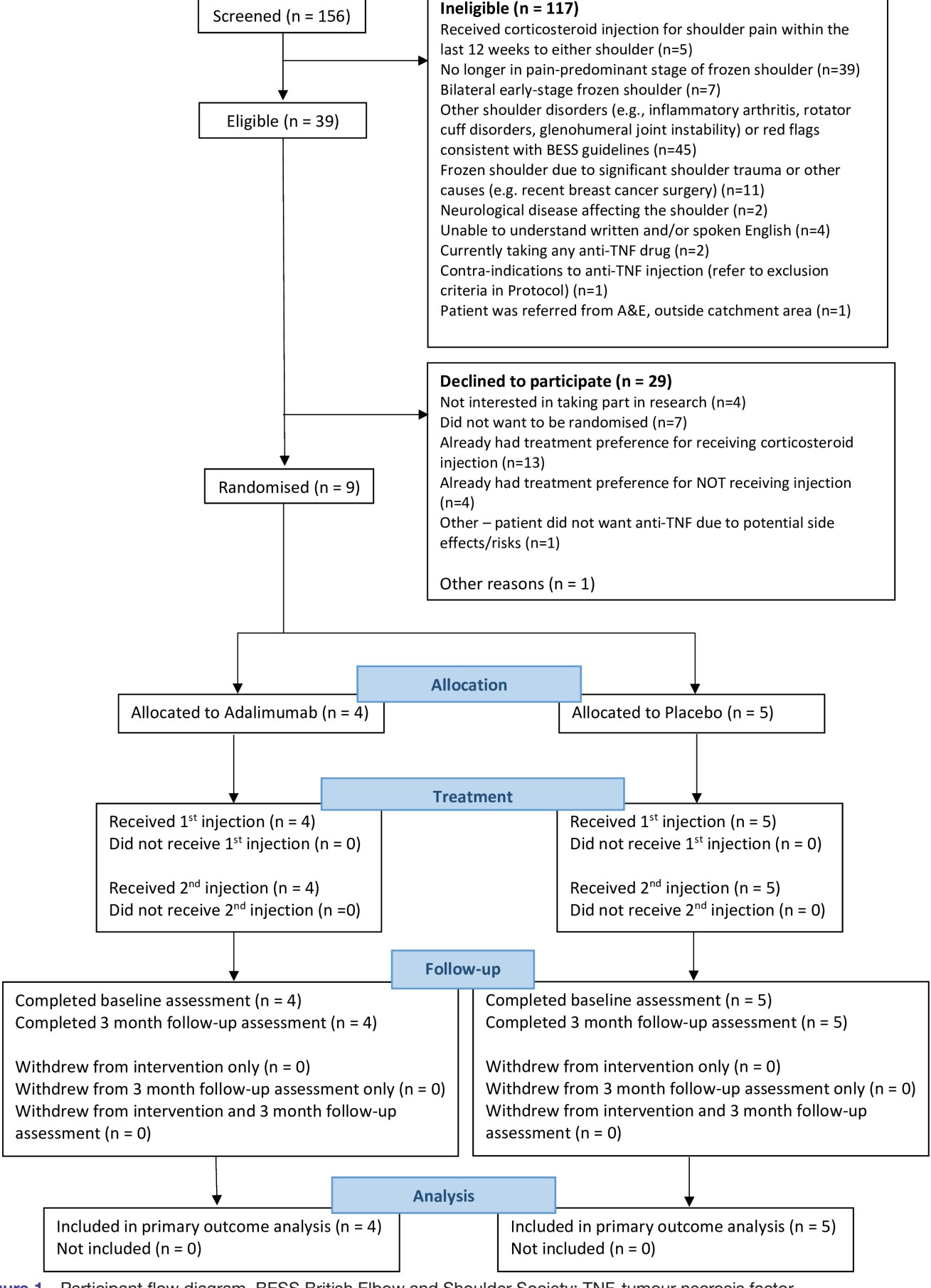

**Figure 1** Participant flow diagram. BESS British Elbow and Shoulder Society; TNF, tumour necrosis factor.

**Table 1** Baseline characteristics by intervention group

|  | Adalimumab (n=4) | Placebo (n=5) | Total (n=9) |
|---|---|---|---|
| Age* | 57.5 (5.9) | 52.8 (5.0) | 54.9 (5.6) |
| Sex† |  |  |  |
| Male | 3 (75.0%) | 3 (60.0%) | 6 (66.7%) |
| Female | 1 (25.0%) | 2 (40.0%) | 3 (33.3%) |
| Ethnicity† |  |  |  |
| White British | 4 (100.0%) | 5 (100.0%) | 9 (100.0%) |
| Handedness† |  |  |  |
| Left | 2 (50.0%) | 1 (20.0%) | 3 (33.3%) |
| Right | 2 (50.0%) | 4 (80.0%) | 6 (66.7%) |
| Duration of symptoms (weeks)* | 19.5 (13.1) | 12.2 (2.3) | 15.4 (9.0) |
| Affected shoulder† |  |  |  |
| Left | 3 (75.0%) | 2 (40.0%) | 5 (55.6%) |
| Right | 1 (25.0%) | 3 (60.0%) | 4 (44.4%) |
| BMI (kg/m$^2$)* | 29.5 (5.8) | 30.3 (5.3) | 30.0 (5.2) |
| Smoking status† |  |  |  |
| Never smoked | 4 (100.0%) | 3 (60.0%) | 7 (77.8%) |
| Former smoker | 0 | 2 (40.0%) | 2 (22.2%) |
| If ever smoked, number of cigarettes/day* | 0 | 12 (11.3) | 12 (11.3) |
| Dupuytren's disease† |  |  |  |
| No | 4 (100.0%) | 4 (80.0%) | 8 (88.9%) |
| Yes | 0 | 1 (20.0%) | 1 (11.1%) |
| Diabetes† |  |  |  |
| Type 1 | 0 | 1 (20.0%) | 1 (11.1%) |
| Type 2 | 1 (25.0%) | 2 (40.0%) | 3 (33.3%) |
| No | 3 (75.0%) | 2 (40.0%) | 5 (55.6%) |

\* Summaries are mean (SD).
†Summaries are n (%).

**Table 2** Injection delivery

|  | Adalimumab (n=4) | Placebo (n=5) | Overall (n=9) |
|---|---|---|---|
| First injection given* | 4 (100%) | 5 (100%) | 9 (100%) |
| Time from randomisation to first injection (days)† | 12.3 (4.8) | 7.2 (4.1) | 9.4 (4.9) |
| Injection administered?* |  |  |  |
| Yes—fully | 4 (100%) | 5 (100%) | 9 (100%) |
| Second injection given* | 4 (100%) | 5 (100%) | 9 (100%) |
| Time from first to second injection (days)† | 18.5 (3.3) | 20.8 (5.9) | 19.8 (4.8) |
| Injection administered?* |  |  |  |
| Yes—fully | 4 (100%) | 5 (100%) | 9 (100%) |

*Summaries are n (%).
†Summaries are mean (SD).

at study sites and reported an overall positive experience of participating in the AFF trial.

### Perceived effects of adalimumab and placebo injections

Two participants who received adalimumab injections reported improvements in pain and/or shoulder movements. Another participant reported some pain relief after the first injection. Pain remained unchanged and movement decreased after the second injection. One participant who had received first injection only reported improvement in movement. Two participants who received placebo injections reported increased pain after their first injection that persisted after the second injection. Of the two participants who had only their first injection, one reported noticeable improvement in pain and movement; the other reported increased pain. Two participants (one from each group) were concerned about the

increased pain after the injections. Other concerns were expectations about the long-term effects of adalimumab and continued access to physiotherapy within the trial.

### DISCUSSION

Our findings show that a definitive trial to assess whether an intra-articular injection of adalimumab (anti-TNF) can reduce pain and improve function in people with early-stage frozen shoulder was not feasible within the NHS musculoskeletal and related physiotherapy services that participated in the AFF trial. Only a quarter of the people screened were eligible to take part in this feasibility study; of those eligible, only nine consented to be randomised. Key themes from the semistructured interviews provided a useful insight into the trial and focused on the participant experiences during the pain-dominant stage of frozen shoulder and their perceptions about trial participation, trial processes and outcomes following injection.

One of the main reasons people were ineligible to take part in the AFF trial was because they were no longer in the pain-predominant phase of frozen shoulder by the time they were seen by the physiotherapist. People no longer being in the pain-predominant phase of frozen shoulder is indicative of long waiting lists and time taken to be seen across NHS musculoskeletal and related physiotherapy services and was a major barrier in terms of trial feasibility. Long waiting times is a problem across the NHS, particularly during and after the COVID-19 pandemic;[33] the period in which the AFF trial was recruiting and not specific to people with frozen shoulder. The four NHS sites that participated in the AFF trial represented several different musculoskeletal and physiotherapy service delivery approaches, with approaches also changing during and post the COVID-19 pandemic. Several of these sites also included a small number of First Contact Practitioners (FCPs) embedded with primary care.

**Table 3** Themes and participant quotes from qualitative interviews

| Themes | Narrative quotes |
| --- | --- |
| Experiences with pain-predominant phase of frozen shoulder | ▶ 'I always put childbirth as a 10, so I always come back from that really, so I would say it was kind of like an eight to nine (Adalimumab)<br>▶ 'It affects when I pretty much … every day, whatever I'm doing' (Placebo)<br>▶ ' it just doesn't help me sleeping. I'm getting about three hours and max four hours at nights…' (Placebo) |
| Perceptions about participation in the trial | ▶ '…if it's gonna help me and help other people then, yeah, it's something I'd do' (Placebo)<br>▶ 'My first thought was yes, I was very happy to, primarily because the current treatment hadn't been particularly successful and I was in still quite a bit of pain and struggling to sleep very well and I thought the fact that another option was being made available, I was more than comfortable to participate' (Adalimumab)<br>▶ 'my expectation really was that if I did get the drug it was likely that I would see some kind of improvement' (Adalimumab)<br>▶ 'I hoped that I got the drug and not the placebo and it helped' (Adalimumab)<br>▶ 'I accepted that that's now how drug trials work and, therefore, I knew there was a possibility that I may not be receiving the drug' (Adalimumab)<br>▶ 'I mean initially I was concerned because if I was the placebo I felt that potentially I was gonna be, well maybe three months further down the line having had the placebo and no benefit. And then I'd have to go back to square one so I was slightly anxious about that. But I was reassured that it, you know, it wasn't stopping me then getting further treatment if I hadn't had any response to the injections. And I guess, this is a bit sceptical of me that waiting lists are what they are but I probably would have been waiting three months for a steroid injection anyway. So, I didn't think it as a disadvantage to do the trial really. If it worked, great, and if it didn't, then I'd-- like I say, I'd go back to the GP' (Placebo) |
| Perceptions about trial processes | ▶ 'I couldn't fault the information I was given to be honest, they were very good' (Placebo)<br>▶ 'I just went in had the injection. It wasn't very pleasant. It was quite…quite painful. It did make me feel a little bit queasy at the time' (Adalimumab)<br>▶ 'I was happy to do that because I understand that they have to gauge any improvement in my shoulder movements over time. So, yeah, I was happy to complete those movements' (Placebo)<br>▶ 'I mean it all went well. I was seen on time, everything was ready. Everything went to plan' (Adalimumab)<br>▶ 'I'm really delighted that I'm part of the trial now because I'm seeing the benefit of it already. So, I'm very … I'm thrilled. I feel very lucky… I mean I do feel-- I feel honoured and I feel like they've treated me so well' (Placebo) |
| Perceptions about effects of adalimumab and placebo injections | After first and second adalimumab injections<br>▶ 'I think there was a noticeable improvement from the first injection. Perhaps a slightly… bit of an extra improvement after the second one. But I think it was quite noticeable following that first injection' (Adalimumab)<br>▶ Perhaps a slightly… bit of an extra improvement after the second one. But I think it was quite noticeable following that first injection'. I definitely have got significant reduction in the pain. I've got more what I would say is a tightness in the muscular areas now, so this almost feels as though where I was on the old regime – pre-trial' (Adalimumab)<br>After first and second placebo injections<br>▶ 'once I had the first injection, my arm … the actual … the pain actually increased quite significantly in my shoulder for a good 7 to 10 days. And I was having to take strong painkillers to sort of manage that. It's actually settled down a bit now' (Placebo)<br>▶ '…there wasn't any significant increase in pain after the second injection…I've not noticed any real improvement since I had the second injection…I still do get the discomfort that I've got in my shoulder' (Placebo) |

However, given the very small number of trial participants and small number of FCPs, we are not able to determine whether moving intervention delivery earlier in the patient pathway would have improved recruitment. In addition, our early feasibility work to identify potential trial sites identified lack of local site pharmacy (ie, for storage and dispensing of adalimumab) and delivery of injection under ultrasound guidance was a major barrier to conducting the AFF trial in primary care and should be considered for any future trial.

We employed a number of different recruitment strategies to try and maximise recruitment at sites. This included sending targeted trial information packs to GPs and healthcare professionals, such as FCPs, within a 15 mile radius of each recruiting site. We promoted the trial at scientific conferences targeting clinicians in the

shoulder community and created a short trial promotional video for the Chartered Society of Physiotherapists. We also promoted the trial on social media with input from our patient representative, and through patient organisations such as the Dupuytren's Society and Diabetes UK to target groups in whom the incidence of frozen shoulder is higher.[1 13]

Another main barrier to feasibility of the trial was because some people, who were eligible and approached to take part in the AFF trial, already had a treatment preference for receiving a steroid injection and declined to participate. We did consider the use of corticosteroid injection as opposed to placebo when designing the trial; however, given the available evidence showing that steroid injection only leads to short-term benefit,[8 9] we decided to use a placebo injection of saline (0.9% sodium chloride) as the comparator. Unlike steroid (triamcinolone), which is a white suspension, saline has the same appearance to adalimumab meant we were able to blind participants as to their treatment allocation. This was especially important given the subjective nature of some of the outcomes we planned to assess in the definitive trial, should this study have shown to be feasible. Interestingly, one of the main findings from our interviews was that participants understood the concept of randomisation and accepted the possibility of receiving placebo.

Despite not meeting the feasible objectives in terms of recruitment and success of the consent process, it was feasible to deliver the intervention in line with the trial protocol and all participants completed patient-reported and clinician assessed outcomes at 3 months. Data from the semistructured interviews showed that participants found the overall experience of the trial was positive. They participated in the AFF trial considering it as an opportunity to find a solution to their shoulder problem and also to benefit others.

## Strengths

To our knowledge, AFF is the first study to assess the feasibility of conducting a large multicentre trial to test the effectiveness of a new treatment for people with early stage, pain-predominant frozen shoulder. This is important given the evidence for the limited effectiveness for treatments currently offered to people with frozen shoulder.[7 8 10 11 34] The drug adalimumab has a very strong safety profile, having been used in over 5 million people, more than 25 000 trial participants and approved for nine different disorders.[35] We used a targeted injection approach using ultrasound guidance, which was standardised across sites and meant we were able to maintain blinding of trial participants and members of trial team at sites. Despite the small number of participants recruited into the trial, our interviews provided a useful insight into patient experiences with the early painful phase of frozen shoulder and their perceptions about being involved in the AFF trial. AFF is one of the very few feasibility trials of upper limb conditions that include a qualitative component.

## Limitations

A limitation of this study is that all participants were white British with an average age of 55 years. Therefore, the population recruited were not truly representative of the general population or those people affected by frozen shoulder,[36] despite recruiting from sites within inner cities with a more diverse population demographic. In addition, while all nine participants recruited into the trial received both their first and second injection within the specific timeframe, it is unclear whether this would be same for larger definitive trial with more pressure on treatment appointment slots. Similarly, while all participants completed both patient-reported and clinician-assessed outcomes at 3 months, it is unlikely this would be same for larger trial with longer follow-up. In a recent trial of corticosteroid injection and physiotherapy in people with a rotator cuff disorder, carried out across 20 NHS musculoskeletal services, the attrition rate at 6 and 12 months was 13%.[37] It is also uncertain whether data saturation was reached from the eight semistructured interviews and, therefore. These findings may not be generalisable and should be interpreted only within the context of the AFF trial.

## CONCLUSION

Our findings demonstrated that a definitive trial was not feasible within the NHS musculoskeletal and related physiotherapy services that participated in the AFF trial. This was largely due to difficulties in identifying patients while still in early-stage frozen shoulder and patient expectations regarding steroid injection. However, it was feasible to deliver the injection in line with trial protocol and to collect both patient and clinician assessed outcome data. Our interview findings indicated positive experiences from participants who took part in the trial.

**Author affiliations**
[1]Oxford Clinical Trials Research Unit, Nuffield Department of Orthopaedics, Rheumatology and Musculoskeletal Sciences, University of Oxford, Oxford, UK
[2]Centre for Rehabilitation Research in Oxford, Nuffield Department of Orthopaedics, Rheumatology and Musculoskeletal Sciences, University of Oxford, Oxford, UK
[3]College of Medicine and Health, University of Exeter, Exeter, UK
[4]Kennedy Institute of Rheumatology, Nuffield Department of Orthopaedics, Rheumatology and Musculoskeletal Sciences, University of Oxford, Oxford, UK
[5]South Tees Hospitals NHS Foundation Trust, The James Cook University Hospital, Middlesbrough, UK
[6]Department of Health Sciences, University of York, York, UK
[7]Patient and Public Involvement Representative, Kidderminster, UK

**Acknowledgements** This study is coordinated by the UKCRC registered Oxford Clinical Trials Research Unit (OCTRU) at the University of Oxford. It followed their Standard Operating Procedures, ensuring compliance with the principles of Good Clinical Practice and the Declaration of Helsinki and any applicable regulatory requirements. Special thanks go Nicola Kenealy as trial manager and Sue Dutton as co-applicant (now retired) and members of the Trial Oversight Committee: Ben Fisher (chair; University of Birmingham), Jim Wiggle (Patient Representative), Rebecca Evans (University of Bristol), Mike Thomas (Heatherwood and Wexham Park NHS Hospitals Trust), Anju Jaggi (Royal National Orthopaedic Hospital NHS Trust).

**Collaborators** The Anti-Freaze-F study collaborating principal investigators were Vincent Gallagher (East Sussex Healthcare NHS Trust), Neil Smith (Sandwell and West Birmingham Hospital NHS Trust), Alun Yewlett (United Lincolnshire Hospitals NHS Trust), Paul Barker (University Hospitals Birmingham).

**Contributors** SH and JN are co-chief investigators and co-lead applicants. AR, MF, JP and SL are co-applicants on the grant awarded by the National Institute of Health Research—Research for Patient Benefit Programme (Project reference: 201031) and were involved in design and conduct of the study, and interpretation of data. CS led the qualitative study, FE and SM conducted the statistical analysis and AE and EAF provided trial management expertise. SH, JN and CS led on writing the manuscript. All authors read, commented and approved the final version. SH is responsible for the overall content as guarantor and accepts full responsibility for the study, had access to the data and controlled the decision to publish.

**Funding** This report is independent research funded by the National Institute of Health Research—Research for Patient Benefit Programme (NIHR: 201031). The views expressed in this publication are those of the author(s) and not necessarily those of the NIHR or the Department of Health and Social Care. 180 Life Sciences provided funding for the purchase of adalimumab, the investigational medicinal product used in this trial.

**Disclaimer** The views expressed are those of the author(s) and not necessarily those of the NHS, the NIHR or the Department of Health.

**Competing interests** MF and JN are co-founders and hold equity in 180 Life Sciences, the company who funded the purchase of adalimumab, the investigational medicinal product used in this trial.

**Patient and public involvement** Patients and/or the public were involved in the design, or conduct, or reporting, or dissemination plans of this research. Refer to the Methods section for further details.

**Patient consent for publication** Not applicable.

**Ethics approval** This study involves human participants and was approved by Health Research Authority North East - Newcastle & North Tyneside 1 (REC 21/NE/0214). Participants gave informed consent to participate in the study before taking part.

**Provenance and peer review** Not commissioned; externally peer reviewed.

**Data availability statement** Data are available upon reasonable request. Summary results data will be available on the trial registration databases within 12 months of the end of the trial. Requests for data (anonymised trial participant level data) will be provided at the end of the trial to external researchers who provide a methodologically sound proposal to the trial team (and who will be required to sign a data sharing access agreement with the Sponsor) and in accordance with the NIHR guidance. Participant consent for this was included in the informed consent form for the study.

**ORCID iDs**
Sally Hopewell http://orcid.org/0000-0002-6881-6984
Amar Rangan http://orcid.org/0000-0002-5452-8578
M Sofia Massa http://orcid.org/0000-0003-3205-6706

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
