## [Reviewer comments · BMJ Open]

ARTICLE DETAILS

TITLE (PROVISIONAL)	Anti-TNF (adalimumab) injection for the treatment of pain predominant early-stage frozen shoulder: the Anti-Freeze-Feasibility randomised controlled trial.
AUTHORS	Hopewell, Sally; Srikesavan, Cynthia; Evans, Alison; Er, Fema; Rangan, Amar; Preece, Jane; Francis, Anne; Massa, M Sofia; Feldmann, Marc; Lamb, Sarah; Nanchahal, Jagdeep

VERSION 1 – REVIEW

REVIEWER	Kim, Du Hwan Chung-Ang University
REVIEW RETURNED	05-Oct-2023

GENERAL COMMENTS	The authors conducted the AFF trial to assess the feasibility of intra-articular injection of adalimumab (anti-TNF) in people with early-stage frozen shoulder. Despite the disappointing results of this study, in which only a small number of participants were enrolled, it is encouraging that the authors described the interpretation of these results well and richly described the results through in-depth interviews conducted on selected patients. However, I have a few questions. 1. According to the authors' previous protocol article, the usefulness of adalimumab in dupuytren contracture is explained as the background to this study. However, dupuytren contracture and frozen shoulder are clinically considered to be completely different diseases. Although the pathophysiology is similar, the two diseases are understood to be different in terms of their natural history. In the early-stage frozen shoulder, intraarticular steroid injection treatment is already considered a cost-effective treatment. The background for the application of adalimumab to frozen shoulder should be provided.2. According to the protocol, the authors defined "frozen shoulder" based on the criteria of BESS. Nevertheless, in clinical practice, it is difficult to differentiate between frozen shoulder and mimicking diseases, especially rotator cuff stiffness. It is necessary to describe the frequency of use of MRI and US in addition to x-ray examination for differential diagnosis of frozen shoulder from other mimicking diseases. Additionally, the definition of early stage FS is not accurately described in this paper. Of course, it is a well-known fact that early stage FS and stiffness predominant FS cannot be distinguished like a razor cut, but defining early stage FS seems to be an important task when designing a study.3. There is no exact explanation of how ROM is measured. The results of ROM vary greatly depending on how it is measured. For example, it varies depending on whether the measurement was performed sitting or lying down, whether scapular fixation or
---

	scapular movement was allowed, and whether the angle of external or internal rotation was measured in abducted or adducted condition. 4. How was the success of intra-articular injection confirmed? Does ultrasound-guided injection guarantee 100% success? Is there a possibility that only part of it entered the joint space? 5. What is the basis for determining the dose of adalimumab as the loading dose used for inflammatory bowel disease? Is it reasonable to set the doses of subcutaneous injection and intra-articular injection as equivalent?
--	---

REVIEWER	Guo, Jiong Jiong Soochow University Affiliated No 1 People's Hospital, Orthopedics and Sports Medicine
REVIEW RETURNED	27-Dec-2023

GENERAL COMMENTS	 1. Regarding the diagnostic criteria for frozen shoulder, since it is mentioned to exclude shoulder joint diseases such as rotator cuff injury, should ultrasound or magnetic resonance imaging be performed? If it is intended to reflect real-world-practice, a physical examination of the shoulder should be performed. The paper only mentioned 'Imaging, including plain radiographs' to 'confirm the diagnosis of frozen shoulder and rule out other...', and such a statement seems to be not rigorous. 2. The author said randomisation was computer-generated and stratified by study site using a variable block size. Can you further explain the size of the block. 3. The design of this study is double-blind, and whether the patient is blind to the treatment plan, please further explain in the "Blind" section. Meanwhile the authors may consider using the James Blinding Index to assess the success of blinding. 4. The range of shoulder movement in this study was assessed by only one assessor, which may affect the accuracy of the assessment. Can the authors further explain how to improve the accuracy of the assessment, such as the double assessment method? 5. In "Informed consent, baseline assessment and trial specific screening tests", it is best to indicate the reason for the serological testing to gain better understandability and avoid unnecessary misunderstandings, just as mentioned in response that continuous administration of anti-TNF is associated with reactivation of latent TB or hepatitis B. 6. In "Figure 1", related information of "Embedded qualitative study", such as numbers and time point, may be better to be added. This could make the study flow of protocol more complete and comprehensive. 7. In "Objectives" and "outcomes - Feasibility objectives", the content of two parts seems to be a little repetitive although they are structurally necessary. Despite this, I wonder whether it is better to add the fourth point "standard deviation for a definitive trial" in "Objectives" as a secondary objective. 8. This version removes "strengths" and "limitations" of the original version. I wonder why this deletion was made. It is necessary to summarize the shortcomings and draw a similar conclusion in discussion section.
---

VERSION 1 – AUTHOR RESPONSE

Reviewer: 1

1. According to the authors' previous protocol article, the usefulness of adalimumab in Dupuytren contracture is explained as the background to this study. However, Dupuytren contracture and frozen shoulder are clinically considered to be completely different diseases. Although the pathophysiology is similar, the two diseases are understood to be different in terms of their natural history. In the early-stage frozen shoulder, intraarticular steroid injection treatment is already considered a cost-effective treatment. The background for the application of adalimumab to frozen shoulder should be provided. RESPONSE: We have strengthened the introduction to provide more detail on the background of the application of adalimumab to frozen shoulder and evidence showing a lack of long term benefit for intraarticular steroid injection. Dupuytren's disease has a prominent genetic component and in a twin study the heritability was estimated at 80% (Larsen J Hand Surg 2015, 40E: 171 -176). We have recently confirmed that there is a significant genetic correlation between Dupuytren's disease and frozen shoulder (Riesmeijer Nature Comms 2024, 15: 199). Approximately 50 % of patients with frozen shoulder also develop Dupuytren's disease (Smith 2001). The pathology of the two conditions has several similarities (Bunker 1995, Smith 2001). A systematic review of the pathophysiology of frozen shoulder identified the key role of inflammation (Ryan 2016) and several inflammatory cytokines, including TNF have been implicated (Bunker 2000, Lho 2013). We recently completed a phase 2b clinical trial of intralesional injections of adalimumab in patients with early stage Dupuytren's disease and met the primary and key secondary endpoint (Nanchahal Lancet Rheum 2022).

Whilst we agree intra-articular steroid injections are often administered to patients with early stage frozen shoulder, there are limited data to support their use. Two Cochrane reviews have concluded that whilst oral they lead to short term benefit in pain and range of motion, the effects are not maintained beyond 6 weeks. A more recent systematic review (based on data from 5 RCTs) also found some short-term benefit of corticosteroid injection compared to placebo but this pain relief was not sustained in the long term.

2. According to the protocol, the authors defined "frozen shoulder" based on the criteria of BESS. Nevertheless, in clinical practice, it is difficult to differentiate between frozen shoulder and mimicking diseases, especially rotator cuff stiffness. It is necessary to describe the frequency of use of MRI and US in addition to x-ray examination for differential diagnosis of frozen shoulder from other mimicking diseases. Additionally, the definition of early stage FS is not accurately described in this paper. Of course, it is a well-known fact that early stage FS and stiffness predominant FS cannot be distinguished like a razor cut, but defining early stage FS seems to be an important task when designing a study.

RESPONSE: Early stage frozen shoulder (i.e. within approximately three to nine months of onset of symptoms) was diagnosed clinically using the criteria set out in the BESS guidelines. In the NHS in the UK imaging is not routinely used to diagnose frozen shoulder, rather it may be used to exclude other suspected shoulder pathologies such as glenohumeral joint arthritis. Therefore, we did not record whether any participants had received imaging prior to being consented to take part in this study. We have added more detailed explanation on the use of imaging to the methods section of the manuscript.

3. There is no exact explanation of how ROM is measured. The results of ROM vary greatly depending on how it is measured. For example, it varies depending on whether the measurement was performed sitting or lying down, whether scapular fixation or scapular movement was allowed, and whether the angle of external or internal rotation was measured in abducted or adducted condition.

RESPONSE: ROM was measured by a blinded assessor using a manual goniometer. We did not specify whether the measurement was performed sitting or lying down etc. Given the pragmatic nature of the future definitive trial we allowed clinicians for use their own clinical judgment when performing this assessment. The same assessor measured ROM at baseline and at 3 months, thus any variability would be consistent within the same person. We agree greater standardisation would have been more rigorous. However, as this was a feasibility study this was not one of our main feasibility objectives.

4. How was the success of intra-articular injection confirmed? Does ultrasound-guided injection guarantee 100% success? Is there a possibility that only part of it entered the joint space?

RESPONSE: Injection details, including the success of the intra-articular injection (i.e. fully or partial), was recorded on a trial specific injection treatment log. We have added this to the methods section of the manuscript. If the injecting clinician recorded partial on the injection treatment log then they were asked to record the estimated volume which was received. Of the 9 participants included in the study all injections were recorded as fully injected.

5. What is the basis for determining the dose of adalimumab as the loading dose used for inflammatory bowel disease? Is it reasonable to set the doses of subcutaneous injection and intra-articular injection as equivalent?

RESPONSE: The loading dose of adalimumab for inflammatory bowel disease (160mg followed by 80mg 2 weeks later) is the highest approved use. Whilst we accept that subcutaneous administration is not equivalent to intra-articular injection, it is likely that any potential systemic adverse effects would be similar between the two routes of administration. Further, as the half-life of adalimumab is 2 weeks administration of the second dose at this interval would be most likely to achieve optimal therapeutic levels. We administered the adalimumab at the site of maximal inflammation using an anterior approach under ultrasound guidance to ensure maximal efficacy.

Reviewer: 2

1. Regarding the diagnostic criteria for frozen shoulder, since it is mentioned to exclude shoulder joint diseases such as rotator cuff injury, should ultrasound or magnetic resonance imaging be performed? If it is intended to reflect real-world-practice, a physical examination of the shoulder should be performed. The paper only mentioned 'Imaging, including plain radiographs' to 'confirm the diagnosis of frozen shoulder and rule out other...', and such a statement seems to be not rigorous.

RESPONSE: Detailed inclusion and exclusion criteria are provided in the published trial protocol. In the NHS in the UK, imaging such as ultrasound or MRI is not routinely used to diagnosis frozen shoulder. As per standard NHS care imaging was only to be used to rule out other shoulder conditions as opposed to confirming a diagnosis of frozen shoulder. We agree a physical examination of the shoulder should be used to confirm diagnosis of frozen shoulder in real world practice and is part of the BESS guideline used to confirm eligibility in this trial. We have clarified this.

2. The author said randomisation was computer-generated and stratified by study site using a variable block size. Can you further explain the size of the block.

RESPONSE: The study used variable block sizes of 2, 4, 6 in a ratio 1:2:1, we have added this to the section on randomisation.

3. The design of this study is double-blind, and whether the patient is blind to the treatment plan, please further explain in the "Blind" section. Meanwhile the authors may consider using the James Blinding Index to assess the success of blinding.

RESPONSE: In the methods section we state "Study participants and site staff, except pharmacy staff, were blinded to treatment allocation. The clinician delivering the treatment injection was not blinded but was not involved in further trial-specific assessment of the participant.". Further details on

how blinding was maintained is provided under the description of the intervention. We did not attempt to assess the success of blinding as this is no longer recommended in accordance with CONSORT guidelines and the lack of evidence supporting this practice:
<https://www.bmj.com/content/340/bmj.c332>

4. The range of shoulder movement in this study was assessed by only one assessor, which may affect the accuracy of the assessment. Can the authors further explain how to improve the accuracy of the assessment, such as the double assessment method?

RESPONSE: Range of shoulder motion was only assessed by a single assessor. However, the same assessor measure ROM at baseline and at 3 months, thus any variability would be consistent within the same person. We agree assessment by two independent assessors would have been more rigorous however as this was a feasibility study this was not one of our main feasibility objectives.

5. In “Informed consent, baseline assessment and trial specific screening tests”, it is best to indicate the reason for the serological testing to gain better understandability and avoid unnecessary misunderstandings, just as mentioned in response that continuous administration of anti-TNF is associated with reactivation of latent TB or hepatitis B.

RESPONSE: Full details are provided in the published trial protocol, however, for completeness we have also included this explanation in the methods section.

6. In “Figure 1”, related information of “Embedded qualitative study”, such as numbers and time point, may be better to be added. This could make the study flow of protocol more complete and comprehensive.

RESPONSE: The flow diagram is reported in line with the CONSORT extension for pilot and feasibility study, focusing on the main feasibility objective of the trial. Thus we would respectfully prefer not to include this additional information here as it is described elsewhere in the main body of the text.

7. In “Objectives” and “outcomes - Feasibility objectives”, the content of two parts seems to be a little repetitive although they are structurally necessary. Despite this, I wonder whether it is better to add the fourth point “standard deviation for a definitive trial” in “Objectives” as a secondary objective.

RESPONSE: We have deleted the outcomes subheading. We have not included the feasibility objective related to standard deviation for a definitive trial as this is a feasibility objective but not something we can measure as part of feasibility success criteria.

8. This version removes “strengths” and “limitations” of the original version. I wonder why this deletion was made. It is necessary to summarize the shortcomings and draw a similar conclusion in discussion section.

RESPONSE: We are unclear what is meant here as strengths, limitations and conclusions are provided in the discussion section.